# Vinyl cyclopropanes as a unifying platform for enantioselective remote difunctionalization of alkenes

Xiaoyong Du ⓘ, Marc E. Lennon ⓘ, Georgia Kriticou & Cristina Nevado ⓘ ✉

Asymmetric remote difunctionalization of alkenes is a longstanding challenge in synthetic chemistry, offering the potential to install two functional groups simultaneously across distal carbon atoms in a stereocontrolled manner. While ingenious strategies have been devised to achieve this transformation, a general catalytic system for remote, enantioselective hetero-carbofunctionalization *and* dicarbofunctionalization of alkenes has remained elusive. Here, we present a nickel/photoredox dual-catalyzed asymmetric remote 1,5-carbosulfonylation and 1,5-dicarbofunctionalization of vinyl cyclopropanes. This cascade reaction integrates radical addition, C–C bond cleavage, and cross-coupling to functionalize two distal carbon atoms with high enantioselectivity. Our protocol demonstrates broad substrate scope, excellent functional group tolerance, and significant synthetic utility, as evidenced by late-stage functionalization and product derivatization. Our mechanistic investigations support the involvement of a Ni(0)/Ni(I)/Ni(III) catalytic cycle in our system. This work establishes a versatile platform for remote alkene difunctionalization, expanding the toolbox of enantioselective synthetic methods and unlocking new avenues for complex molecule construction.

Being abundant and readily available feedstocks, alkenes have long been recognized as a platform for the formation of C–C and C–X bonds[1]. This important role has only been amplified as intermolecular alkene difunctionalizations have emerged as a powerful and versatile means of generating molecular complexity through the simultaneous formation of two new bonds across the π-system[2–5]. In this context, numerous radical-mediated intermolecular 1,2-difunctionalizations of alkenes have been successfully developed[6–8], accompanied by a handful of asymmetric examples reported by our group[9–11] and others (Fig. 1a)[12–17].

With heightened recent interest in remote functionalization reactions[18–26], alkene difunctionalizations proceeding via radical tandem processes have emerged as a means of redirecting reactivity away from the initial site of reaction and towards a distal position. Several systems involving radical addition to the C = C bond followed by HAT[27–33], ring-closing[34] or strain-release ring-opening[35–38] processes have been designed for remote alkene difunctionalization. Compared

to metal-mediated chain-walking strategies[39–41], the aforementioned strategies present important benefits including milder regimes, optimal regioselectivity, and broader substrate applicability.

Despite the rapid expansion of this field, asymmetric variants have only recently begun to emerge and remain scarce. Because of the high reactivity and instability of the open-shell intermediates participating in these transformations, one of the principal challenges lies in controlling the configuration of the newly-formed stereogenic centers. Research works from Liu[42,43], Wang[44], and our own group[33] have showcased radical relay systems as a possible strategy to solve this problem. However, these examples strongly rely on the use of perfluoroalkyl radical precursors, such as triflyl chloride, Langlois' reagent and BrCF$_2$CO$_2$Et, thereby inherently limiting their synthetic utility (Fig. 1b). The incorporation of other functional groups in these transformations is an underdeveloped pursuit and, to the best of our knowledge, there has been no report to date of a single, unified

Department of Chemistry, University of Zurich, Zurich, Switzerland. ✉e-mail: cristina.nevado@chem.uzh.ch

**a**

**b**

- Asymmetric remote difunctionalization still rare
- Limited to fluoroalkyl radical precursors

**c**

*non-stereodefined*

**This work**: asymmetric remote hetero-carbofunctionalization and dicarbofunctionalization of alkenes

- Mild conditions
- Mechanistic studies
- High yield and enantioselectivity (up to 71% and 99:1 e.r.)
- Wide substrate scope and functional group tolerance (55 examples)

**Fig. 1 | State of the art of alkene difunctionalization. a** Asymmetric difunctionalization of alkenes via a metal-catalyzed radical pathways. **b** Remote difunctionalization of alkenes via a radical tandem approach. **c** Photoredox and nickel dual-catalyzed asymmetric 1,5-carbosulfonylation and 1,5-dicarbofunctionalization of alkenes via a radical addition/C–C bond cleavage/cross-coupling cascade. TM transition metal. R$^F$ fluoroalkyl radical precursors. PC photocatalyst.

catalytic system for remote, asymmetric hetero-carbofunctionalization *and* dicarbofunctionalization of alkenes.

The ring-opening reactions of the vinyl cyclopropane (VCP) motif boast a decades-long history of both intensive mechanistic study and extensive use in organic synthesis[45]. In particular, their ring expansion to cyclopentenes, known eponymously as the vinyl cyclopropane rearrangement, has become an indispensable tool in the total synthesis of natural products[46]. VCPs' widespread use as radical clocks to probe alkene functionalizations has led to recognition that the cyclopropane ring-opening event provides a strong driving force for distal spin-center transposition[47], and VCPs have thereby emerged as a useful platform for remote difunctionalization in their own right[35–38].

Having rapidly become a staple of method development, and despite recent exceptions[48], VCPs have been overlooked in the context of *asymmetric* remote difunctionalization, and early studies employing them as radical clocks in asymmetric transformations have been characterized by low yields, poor enantioselectivities, and very limited scope[14].

Herein, a versatile dual nickel/photoredox catalytic system is described which enables remote asymmetric 1,5-carbosulfonylation and 1,5-dicarbofunctionalization of vinyl cyclopropanes via a radical addition/C–C bond cleavage/cross-coupling cascade (Fig. 1c). In sharp contrast to previous reports, our protocol boasts synthetically useful yields and enantioselectivities for a wide range of substrates. In addition, the utility and applicability of this methodology have been demonstrated by the success of late-stage functionalization of complex molecules and divergent derivatization of the reaction products into other valuable organic compounds. Finally, we present a series of

experimental studies including detailed elucidation of the catalytic cycle at nickel and the origins of the enantioselectivity.

## Results

### Optimization of reaction conditions

Methyl 4-bromobenzoate, sodium benzenesulfinate and **VCP1** were chosen as model reactants to identify the optimal reaction conditions (Table. 1). The reaction occurred smoothly in the presence of NiCl$_2$·DME (10 mol%) and chiral bisimidazoline (BiIM) ligand **L1** (15 mol%), with B$_2$pin$_2$ (0.85 equiv.), 15-crown-5 (9 equiv.), and Ru(bpy)$_3$(PF$_6$)$_2$ (1 mol%) as photocatalyst under blue light irradiation in DME at 0 °C, giving product **1** as a single geometric isomer in 44% yield and with 86:14 e.r. (entry 1). Screening of different chiral ligands showed that BiIM templates (**L1**–**L3**, entries 1–3) were generally more effective than those based on oxazoline motifs (**L5**–**L7**, entries 4–6) with the sterically demanding N-3,5-di-$^t$BuC$_6$H$_3$-$^s$BuBiIM ligand (**L2**) delivering the best yield (48%) and stereocontrol (90:10 e.r., entry 2). The use of DME proved essential for the success of this transformation, as a substantial decrease in enantioselectivity was observed when THF, acetonitrile, or acetone were used instead (entries 7–9). Two-fold dilution of the reaction had a slight beneficial effect on the enantioselectivity (entry 10). Other nickel(II) precursors, such as NiBr$_2$·DME, delivered the product with similar yield and e.r., while Ni(COD)$_2$ led to a dramatically reduced conversion (entries 11 and 12). Interestingly, the use of precomplexed **L2**·NiBr$_2$ allowed **1** to be obtained with further improved yield (51%) and enantioselectivity (95:5 e.r., entry 13). Finally, adjusting the reaction duration (to 48 h) and stoichiometry (to 48 h, and from 9 to 3 equiv. of 15-crown-5 and 0.85 to 0.1 equiv. of B$_2$Pin$_2$) compound **1** could be

## Table 1 | Optimization of reaction conditions[a]

| Entry | [Ni]/L* | Solvent | Product, yield (%) | E/Z | e.r. |
|---|---|---|---|---|---|
| 1 | NiCl$_2$·DME/L1 | DME | 1, 44 | >20/1 | 86:14 |
| 2 | NiCl$_2$·DME/L2 | DME | 1, 48 | >20/1 | 90:10 |
| 3 | NiCl$_2$·DME/L3 | DME | 1, 43 | >20/1 | 89:11 |
| 4 | NiCl$_2$·DME/L5 | DME | 1, 23 | >20/1 | 75:25 |
| 5 | NiCl$_2$·DME/L6 | DME | 1, 12 | >20/1 | 60:40 |
| 6 | NiCl$_2$·DME/L7 | DME | 1, trace | -- | -- |
| 7 | NiCl$_2$·DME/L2 | THF | 1, 50 | >20/1 | 88:12 |
| 8 | NiCl$_2$·DME/L2 | CH$_3$CN | 1, 47 | >20/1 | 84:16 |
| 9 | NiCl$_2$·DME/L2 | Acetone | 1, 55 | >20/1 | 85:15 |
| 10[b] | NiCl$_2$·DME/L2 | DME | 1, 40 | >20/1 | 93:7 |
| 11[b] | NiBr$_2$·DME/L2 | DME | 1, 42 | >20/1 | 93:7 |
| 12[b] | Ni(COD)$_2$ /L2 | DME | 1, trace | -- | -- |
| 13 | L2·NiBr$_2$ | DME | 1, 51 | >20/1 | 95:5 |
| 14[b,c] | L2·NiBr$_2$ | DME | 1, 72 (70)[d] | >20/1 | 95:5 |
| 15[e] | No Ni or No L2 | DME | 1, ND | -- | -- |
| 16 | NiCl$_2$·DME/L2 | THF | 2, 68 | 8:1 | 86:14 |
| 17 | NiBr$_2$·DME/L2 | THF | 2, 73 | 9:1 | 91:9 |
| 18 | NiCl$_2$·Py$_4$/L2 | THF | 2, 67 (62)[d] | 9:1 | 93:7 |
| 19[f] | NiCl$_2$·Py$_4$/L2 | THF | 2, 35 | 8:1 | 93:7 |
| 20 | NiCl$_2$·Py$_4$/L3 | THF | 2, 48 | 8:1 | 96:4 |
| 21 | NiCl$_2$·Py$_4$/L4 | THF | 2, 45 | 9:1 | 91:9 |
| 22[e] | No Ni or No L2 | THF | 2, ND | -- | -- |

L1, R = $^s$Bu, Ar = 3-$^t$Bu-C$_6$H$_3$-
L2, R = $^s$Bu, Ar = 3,5-$^t$Bu$_2$-C$_6$H$_3$-
L3, R = $^i$Pr, Ar = 3,5-$^t$Bu$_2$-C$_6$H$_3$-
L4, R = cy, Ar = 3,5-$^t$Bu$_2$-C$_6$H$_3$-

*ND* not detected, *DME* dimethoxyethane, *THF* tetrahydrofuran.

[a]For remote carbosulfonylation: *tert*-Butyl 2-vinyl cyclopropane-1-carboxylate (**VCP1**, 0.12 mmol), methyl 4-bromobenzoate (0.1 mmol), PhSO$_2$Na (0.1 mmol), nickel precursor (10 mol %), **L*** (15 mol%), Ru(bpy)$_3$(PF$_6$)$_2$ (1 mol %), 15-crown-5 (9 equiv.), B$_2$Pin$_2$ (0.85 equiv.), solvent (0.1 M), 450 nm 34 W EvoluChem LED, 0 °C, N$_2$, 24 h. For remote dicarbofunctionalization: *tert*-Butyl 2-vinyl cyclopropane-1-carboxylate (**VCP1**, 0.2 mmol), methyl 4-bromobenzoate (0.1 mmol), $^t$BuBF$_3$K (0.25 mmol), Nickel precursor (10 mol %), **L*** (15 mol%), Ir[dF(CF$_3$)ppy)]$_2$(bpy)PF$_6$ (3 mol%), THF (0.05 M), 427 nm 45 W Kessil LED, 20 °C, N$_2$, 24 h. $^1$H NMR yield with mesitylene as internal standard. Enantiomeric ratio (e.r.) and *E/Z* ratio were determined by HPLC with a chiral stationary phase.
[b]DME (0.025 M).
[c]15-crown-5 (3 equiv.), B$_2$Pin$_2$ (0.1 equiv.), 48 h.
[d]Isolated yield.
[e]No nickel, no ligand, no PC, or no light.
[f]0 °C.

obtained in 70% isolated yield and with 95:5 e.r. (entry 14). Gratifyingly, this remote difunctionalization could also be extended to incorporate $^t$BuBF$_3$K as a convenient precursor to a tertiary carbon-centered radical. The desired three-component remote dicarbofunctionalization product **2** was obtained in 62% isolated yield with 93:7 e.r. in the presence of NiCl$_2$·Py$_4$, **L2**, and Ir[dF(CF$_3$)ppy)]$_2$(bpy)PF$_6$ in THF under 45 W, 427 nm LED irradiation at 20 °C for 24 h (entry 18). Employing other nickel(II) precursors or lowering the temperature to 0 °C had no beneficial effect on the enantioselectivity (entries 16–19). Other BiIM ligands, such as **L3**, led to an improvement of the enantioselectivity from 93:7 to 96:4 e.r. but decreased the yield (entries 20–21). Control experiments further

confirmed that the nickel, ligand, photocatalyst, and light were all essential to a successful reaction outcome in both protocols (entries 15 and 22).

## Substrate scope

With the optimized conditions in hand, the scope of the Ni/photo-redox dual-catalyzed remote asymmetric difunctionalization was investigated, beginning with the carbosulfonylation protocol (Fig. 2). We initially focused on evaluating a diverse array of aryl and alkenyl bromides. Functional group classes including esters, nitriles, ketones, halides, and extended ring systems were compatible with the redox-

neutral reaction conditions, giving the corresponding remote carbo-sulfonylation products **1**, **3**–**12** in good yields and with moderate to excellent enantioselectivity. Heteroaryl bromides incorporating dihydroisobenzofuran (**13**), 2-phenylpyridine (**14**), quinoline (**15**), and 1,3-dioxoisoindoline (**16**) moieties, were successfully applied to this synergistic protocol. Moreover, these transformations displayed excellent chemoselectivity towards bromides in the presence of chlorides (**10**, **11**), paving the way for subsequent synthetic manipulation of the products. It should be noted that our system also tolerates alkenyl bromides, allowing enantioenriched alkenyl sulfones **17** and **18** to be isolated in synthetically useful yields, albeit with moderate enantiomeric ratios.

The scope of vinyl cyclopropanes and sulfinates was evaluated next. In addition to **VCP1**, a series of VCPs with different ester substituents proved to be viable substrates, giving the corresponding remote carbosulfonylation products in high yields and with excellent enantioselectivity (**19**–**22**, up to 93:7 e.r.). Notably, our system also worked well with 1,1-disubstituted alkenes, which are typically challenging substrates in Ni-catalyzed intermolecular difunctionalizations[10,13]. 2-Methyl- and 2-phenyl-substituted vinyl cyclopropanes delivered the desired remote difunctionalization products as single geometric isomers in moderate yields and with high enantioselectivities (**23** and **24**). Trisubstituted cyclopropanes were also subjected to the standard reaction conditions and, to our delight, the corresponding remote arylsulfonylated products (**25** and **26**) could be obtained with yields and e.r. values comparable to those observed with the model substrate, thus highlighting the potential of this transformation to accommodate complex cyclopropane motifs as efficient reaction partners. Our chiral photoredox/nickel system could also be applied to enantioenriched ε-sulfonyl-α-arylated amides and ketones, constituting valuable structural motifs in medicinal chemistry (**27**–**28**, up to 90:10 e.r.). Notably, 1-aryl-2-vinylcyclopropanes could be successfully engaged under the reaction conditions to give the corresponding products **29**–**31** with moderate yields and excellent levels of absolute stereocontrol. Moreover, 1,1-disubstituted alkenes of the same class furnished the desired remote difunctionalization products **32** and **33** as single geometric isomers with excellent yields and enantioselectivities. We were delighted to see that sulfinates with both aryl (tolyl, anisyl) and alkyl (cyclopropyl) substituents were compatible with our reaction conditions, affording the corresponding chiral sulfones **34**–**36** in good yields and with high enantioselectivities.

We next turned our attention to evaluating the scope of carbon-based radical precursors compatible with this remote asymmetric difunctionalization (Fig. 3). Under the optimized conditions, a broad range of aryl bromides bearing ester (**2**), trifluoromethyl (**37**), cyano (**38**), sulfonyl (**39**), formyl (**40**), ketone (**41** and **42**), chloro (**43**) and lactone (**44**) groups were all successfully converted to the corresponding remote dicarbofunctionalization products in good yields and with high *E/Z* ratios and e.r. values. Heterocycles such as pyridine (**45**) were well tolerated under the mild reaction conditions. Different vinyl cyclopropane substrates and alkyl trifluoroborate precursors were also evaluated. A wide range of vinyl cyclopropanes delivered the corresponding products in moderate to good yields with up to 8:1 *E/Z* ratios and high enantioselectivities (**46**–**51**). Substrates bearing a 2-substituted olefin were also amenable to this protocol (**50**). As expected, sterically hindered tertiary linear and cyclic alkyl trifluoroborates were applicable to this remote asymmetric photoredox/nickel system (**52**–**55**, up to 93:7 e.r.). Consistent with the presumably lower energy barriers to reductive elimination for less sterically hindered alkyl radicals, 1° and 2° alkyl trifluoroborates were incompetent in the three-component reaction, instead undergoing the corresponding two-component aryl–alkyl cross-coupling to form alkylarenes as the major by-products under the standard conditions (Supplementary Information, Table S8)[49]. It should be noted that in the case of compound **2**, the e.r. could be increased from 93:7 to 96:4 by

employing **L3** as ligand, albeit with a slight decrease in yield. This modification was similarly effective for other substrates, giving the corresponding remote dicarbofunctionalization products (**42, 44, 48, 52, 54**) in moderate yields with up to 95:5 e.r.

## Synthetic applications

The synthetic utility of this remote asymmetric alkene difunctionalization was evaluated on more structurally challenging partners (Fig. 4a). Our method could be successfully applied to the synthesis of an array of complex scaffolds bearing additional stereocenters with no loss of existing stereochemical information. Aryl bromides derived from estrone (**56**), (−)-menthol (**57** and **58**), cholesterol (**59** and **60**), D-glucose (**61** and **62**) and (+)-α-tocopherol (**63** and **64**) furnished the desired remote three-component coupling products in moderate to good yields with excellent diastereoselectivities. It should be noted that the absolute configuration of the difunctionalized products was unambiguously confirmed by X-ray diffraction analysis of (*R,E*)-**60**. The practicality of this nickel and photoredox dual-catalyzed enantioselective remote multicomponent transformation was further demonstrated through product derivatization (Fig. 4b). Selective reduction of the C = C bond in the presence of Pd-C/H₂ resulted in the formation of saturated derivatives **65** and **66** in quantitative yields and with perfect stereoretention (94:6 and 93:7 e.r., respectively). In addition, *tert*-butyl deprotection was successfully accomplished by treatment with TFA. Subsequent esterification delivered methyl esters **67** and **68** with high yields and enantioselectivities, overcoming the limitation to bulky ester substituents imposed by the protocol. Moreover, mild deprotection followed by amidation furnished *L*-phenylalanine derivatives (**69** and **70**) with high levels of diastereocontrol, demonstrating the potential of this method for rapidly accessing complex amino acid derivatives.

## Mechanistic studies

To characterize the mechanism of this transformation, a series of control experiments were conducted (Fig. 5). When enantiopure **VCP10** was used as substrate, the multicomponent reaction occurred smoothly under the standard conditions, giving the corresponding remote difunctionalization products **1** and **2** in 65 and 55% yield with 95:5 and 93:7 e.r., respectively. Replacing chiral ligand **L2** with dtbbpy under otherwise identical conditions, the products were generated in similar yields but as the corresponding racemates (**1′** and **2′**, Fig. 5a). These results indicate that stereocontrol stems from the chiral nickel complex, and that the absolute configuration of the substrate does not affect the stereochemical outcome of the reaction. Next, control experiments with two different radical inhibitors were conducted. The remote difunctionalization reaction was completely suppressed by the addition of 2,2,6,6-tetramethyl-1-piperidinyloxy (TEMPO, 2.0 equiv.) or butylated hydroxytoluene (BHT, 2.0 equiv.). TEMPO adduct **71** was detected in the reaction mixture by high-resolution mass spectrometry (HR-MS) (Fig. 5b). These results hint at the formation of a sulfonyl/alkyl radical that, upon addition to the vinyl cyclopropane and ring opening, delivers a new carbon-centered radical intermediate that can be intercepted in the presence of the abovementioned radical traps. To interrogate the binding pattern of the active catalyst, a series of asymmetric remote dicarbofunctionalization reactions were conducted, varying the enantiopurity of the ligand (**L3**) under the standard reaction conditions (Fig. 5c). A linear effect was observed on the asymmetric induction, indicating a 1:1 binding pattern between **L3** and Ni in the active catalytic species.

To shed light on the nature of the active nickel species in the reaction, several stoichiometric and catalytic experiments were carried out. First, when Ni(I)-Br complex **72**[50] was used stoichiometrically in the reaction of **VCP1** and sodium benzenesulfinate, the corresponding remote arylsulfonylated product **1′** was obtained in 20% yield. Conversely, using catalytic amounts of complex **72** generated the desired product **1′** in 56% yield. This result implicates Ni(I) species as possible

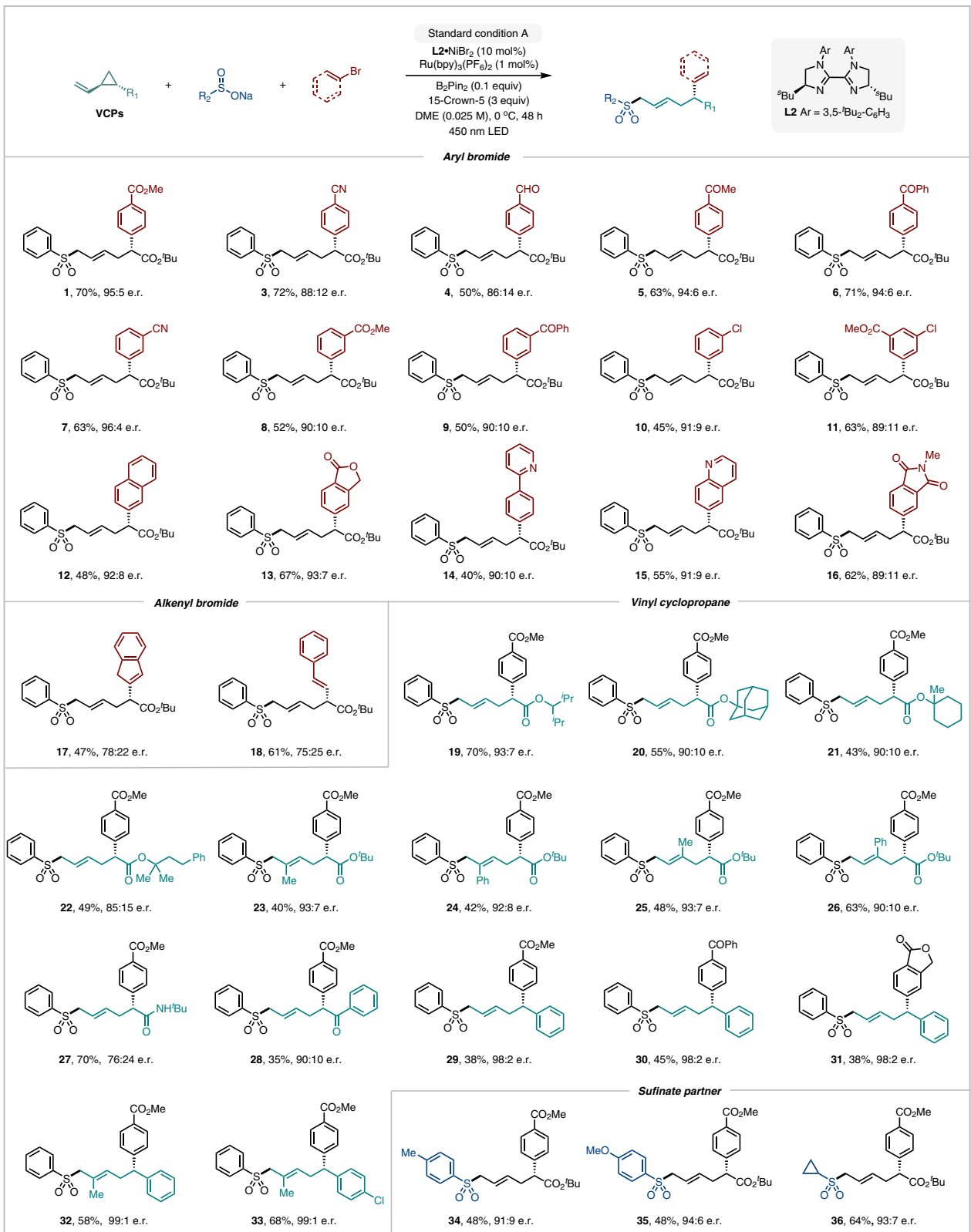

**Fig. 2 | Scope of Ni/photoredox dual catalyzed asymmetric 1,5-carbosulfony-lation of alkenes.** Standard conditions A: vinyl cyclopropane (0.12 mmol), aryl or alkenyl bromide (0.1 mmol), sulfinate precursor (0.1 mmol), **L2**·NiBr₂ (10 mol%), Ru(bpy)₃(PF₆)₂ (1 mol %), 15-crown-5 (3 equiv.), B₂Pin₂ (0.1 mmol), DME (0.025 M), 450 nm 34 W EvoluChem LED, 0 °C, N₂, 24 h. Yield of isolated products, unless noted otherwise. Enantiomeric ratio (e.r.) and (>20:1) E/Z ratio were determined by HPLC with a chiral stationary phase for all products.

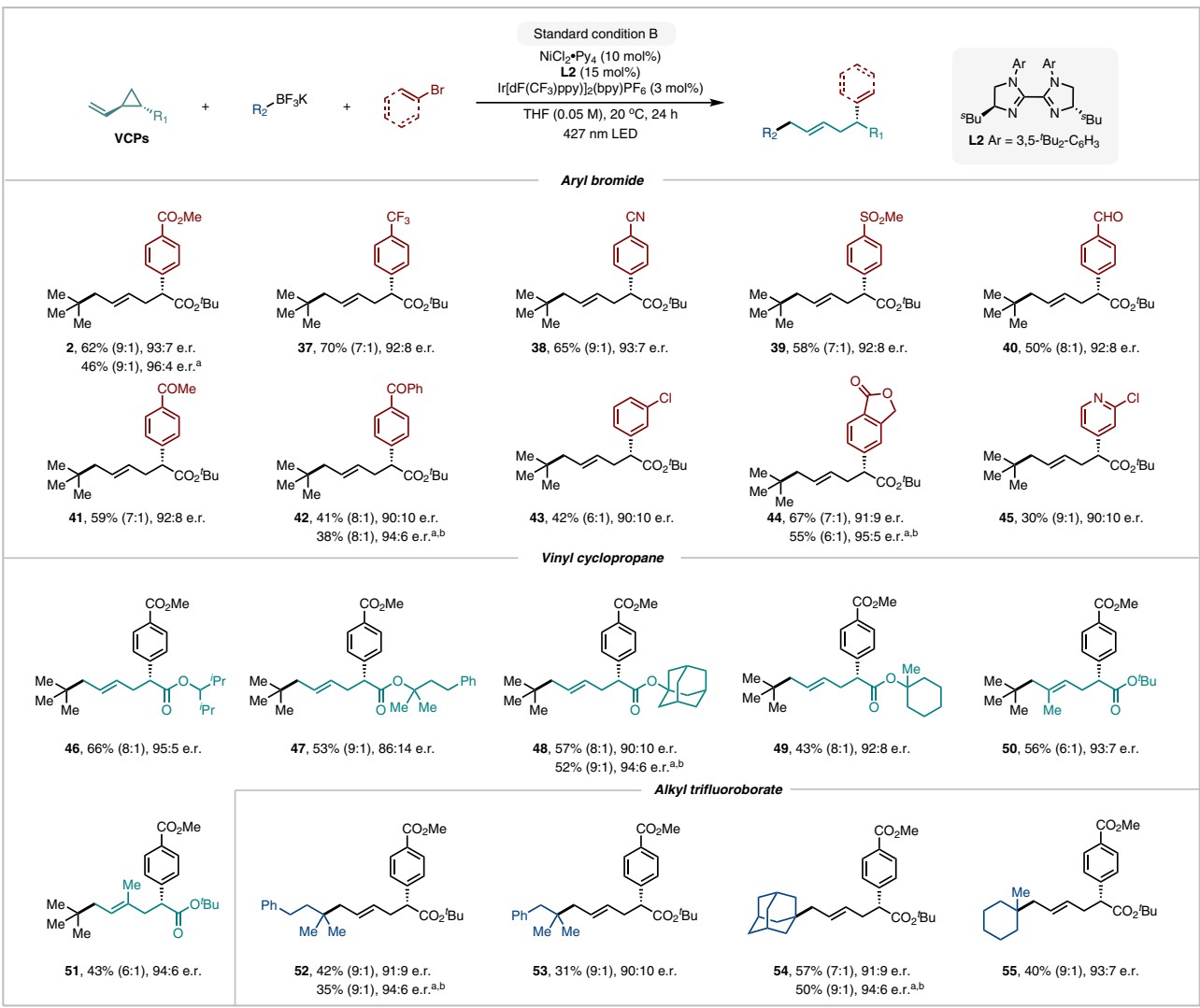

**Fig. 3 | Scope of Ni/photoredox dual catalyzed asymmetric 1,5-dicarbo-functionalization of alkenes.** Standard conditions B: vinyl cyclopropane (0.2 mmol), aryl bromide (0.1 mmol), alkyl trifluoroborate (0.25 mmol), NiCl$_2$·Py$_4$ (10 mol %), **L2** (15 mol%), Ir[dF(CF$_3$)ppy)]$_2$(bpy)PF$_6$ (3 mol%), THF (0.05 M), 427 nm 45 W Kessil LED, 20 °C, N$_2$, 24 h. Yield of isolated products, unless noted otherwise. Enantiomeric ratio (e.r.) and E/Z ratio were determined by HPLC with a chiral stationary phase. a L3 as ligand. b THF (0.025 M).

active catalysts (Fig. 5d). Second, Ar-Ni(II)-Br complex **73** was prepared by oxidative addition of methyl 4-bromobenzoate to Ni(COD)$_2$[10,17], and used in a stoichiometric fashion in the reaction of **VCP1** and sodium benzenesulfinate (Fig. 5e). No conversion to the corresponding remote carbosulfonylation product **1'** could be observed in the reaction mixture. A similar result was also observed in the case of the dicarbofunctionalization protocol. These results reveal that the aryl-Ni(II) species might not be sufficiently reactive to engage with this transformation. In contrast, using catalytic amounts of complex **73** generated the desired products **1'** and **2'** in 75 and 50% yields, respectively (Fig. 5f). Crossover experiments with 4-bromobenzonitrile further supported the catalytic competence of complex **73**, generating the corresponding difunctionalization products **3'** and **38'** in high yields. No **1'** or **2'** was observed in the respective reaction mixtures. Taken together, these experiments suggest that, if formed, Ar-Ni(II) complex **73** serves to generate a low-valent nickel(I) species that is competent in the catalytic cycle.

Based on these results and other recent studies[10,14,51], the following catalytic cycle can be proposed for this nickel/photoredox dual catalyzed remote asymmetric difunctionalization of alkenes (Fig. 5g). First, excitation of the photocatalyst with visible light results in the oxidation of sodium benzenesulfinate or alkyl trifluoroborate by single-electron

transfer (SET) at the expense of the excited catalyst to generate the sulfur- or carbon-centered radical. Radical addition to the C = C bond of the vinyl cyclopropane then produces a secondary alkyl radical **A**, which undergoes rapid ring opening to give a new carbon-centered radical **B**. Intermediate **B** reacts with Ni(0) complex **F** producing (alkyl) Ni(I) species **C**. Upon oxidative addition of the aryl bromide, an (aryl) (alkyl)Ni(III) intermediate **D** is produced. The high-valent Ni(III) intermediate **D** is prone to undergo reductive elimination delivering the remote difunctionalization product and Ni(I)-X species **E**, which is reduced back to complex **F** by the reduced form of the photocatalyst, thus closing the photoredox cycle and regenerating the active nickel catalyst.

## Discussion

A unified means for asymmetric 1,5-carbosulfonylation and 1,5-dicarbofunctionalization of vinyl cyclopropanes is presented here. The synergistic radical addition/C–C bond cleavage/cross-coupling cascade is mediated by a dual photoredox/nickel catalytic system under mild reaction conditions enabling the efficient transformation of readily accessible vinyl cyclopropyl starting materials into a wide spectrum of value-added, remotely difunctionalized products with

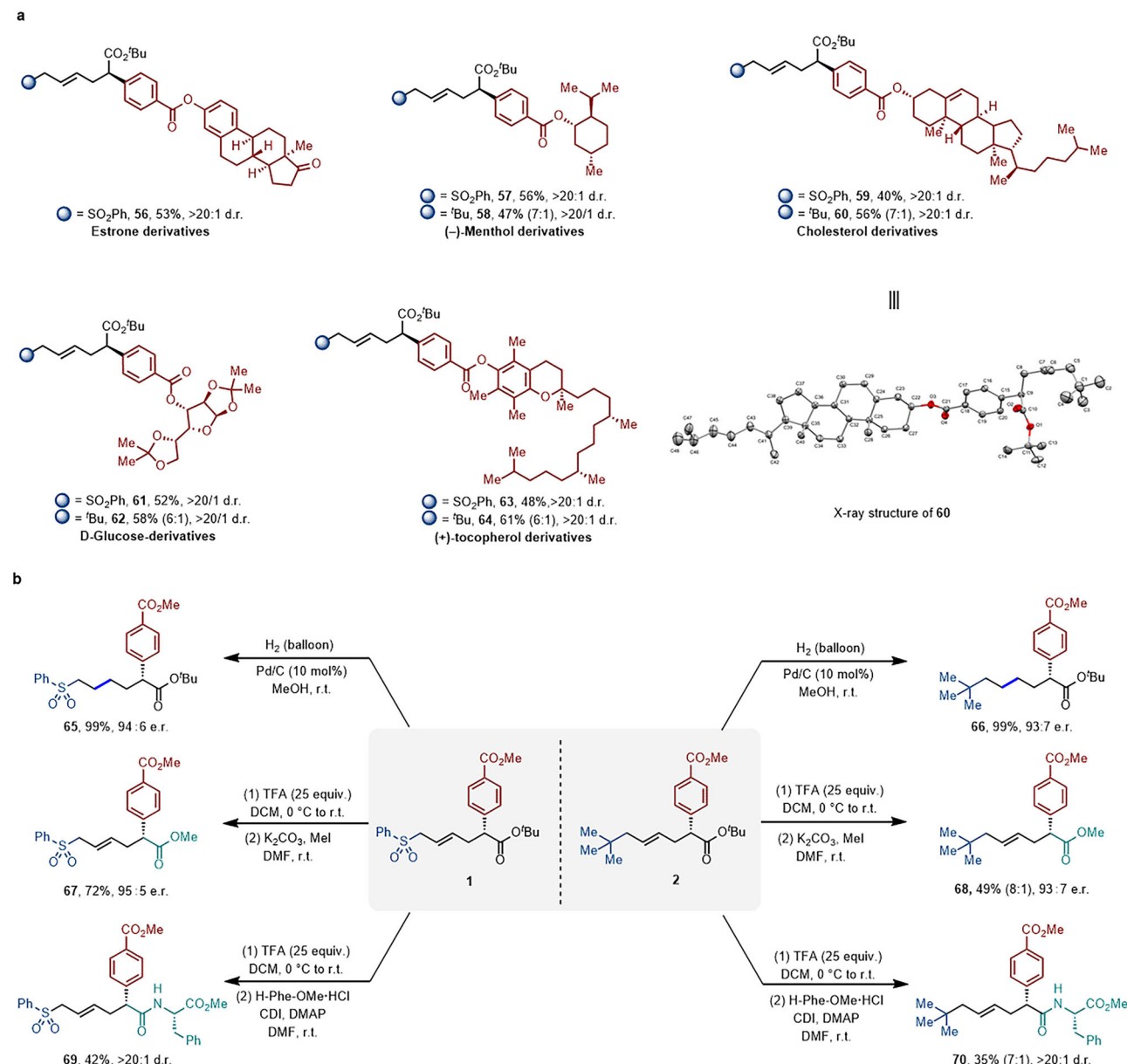

**Fig. 4 | Synthetic applications. a** Late-stage functionalization of complex molecules. **b** Derivatization of products. TFA trifluoroacetic acid, DCM dichloromethane, DMF dimethylformamide, CDI 1,1'-carbonyldiimidazole, DMAP 4-dimethylaminopyridine, H-Phe-OMe·HCl L-phenylalanine methyl ester hydrochloride.

high enantioselectivity. This protocol exhibits high functional group tolerance and has been readily extended to diverse complex molecules, and its synthetic utility was further demonstrated by rapid assembly of biologically relevant scaffolds from the reaction products with no loss of stereochemical information. Mechanistic investigations revealed that the asymmetric induction stems exclusively from the chiral nickel complex capable to recombine with secondary alkyl radical generated after radical addition and ring-opening steps. We are confident that the synthetic methods and mechanistic insights reported here will provide a useful springboard for further studies.

## Methods

### General Procedure for asymmetric 1,5-carbosulfonylation of alkenes (standard conditions A)

An oven-dried 7.5 mL screw-cap vial equipped with a magnetic stirring bar was charged with R$_2$-SO$_2$Na (0.1 mmol, 1 equiv.), aryl bromide or alkenyl bromide (if solid, 0.1 mmol, 1 equiv.), Ru(bpy)$_3$(PF$_6$)$_2$ (1 mol %),

B$_2$Pin$_2$ (0.1 equiv.) and **L2**·NiBr$_2$ complex (0.01 mmol, 10 mol%) and then introduced into a nitrogen-filled glovebox. There, dry DME (4 mL) and 15-crown-5 (0.3 mmol, 3 equiv.) were sequentially added. The reaction vessel was then capped and removed from the glovebox. Vinyl cyclopropane (0.12 mmol, 1.2 equiv.) and aryl or alkenyl bromide (if liquid, 0.1 mmol, 1 equiv.) were subsequently added. The reaction was stirred (800 rpm) under irradiation with a 34 W 450 nm LED at 0 °C for 48 h. The reaction was quenched with saturated aq. NaCl (1 mL) and the resulting mixture were extracted with EtOAc (3 × 2 mL). The organic phase was concentrated under reduced pressure, and the residue purified by column chromatography on silica gel. The enantiomeric ratio (e.r.) and *E/Z* ratio of the product were determined by HPLC with a chiral stationary phase.

### General procedure for asymmetric 1,5-dicarbofunctionalization of alkenes (standard conditions B)

An oven-dried 7.5 mL screw-cap vial equipped with a magnetic stirring bar was charged with R$_2$-BF$_3$K (0.25 mmol, 2.5 equiv.), aryl bromide

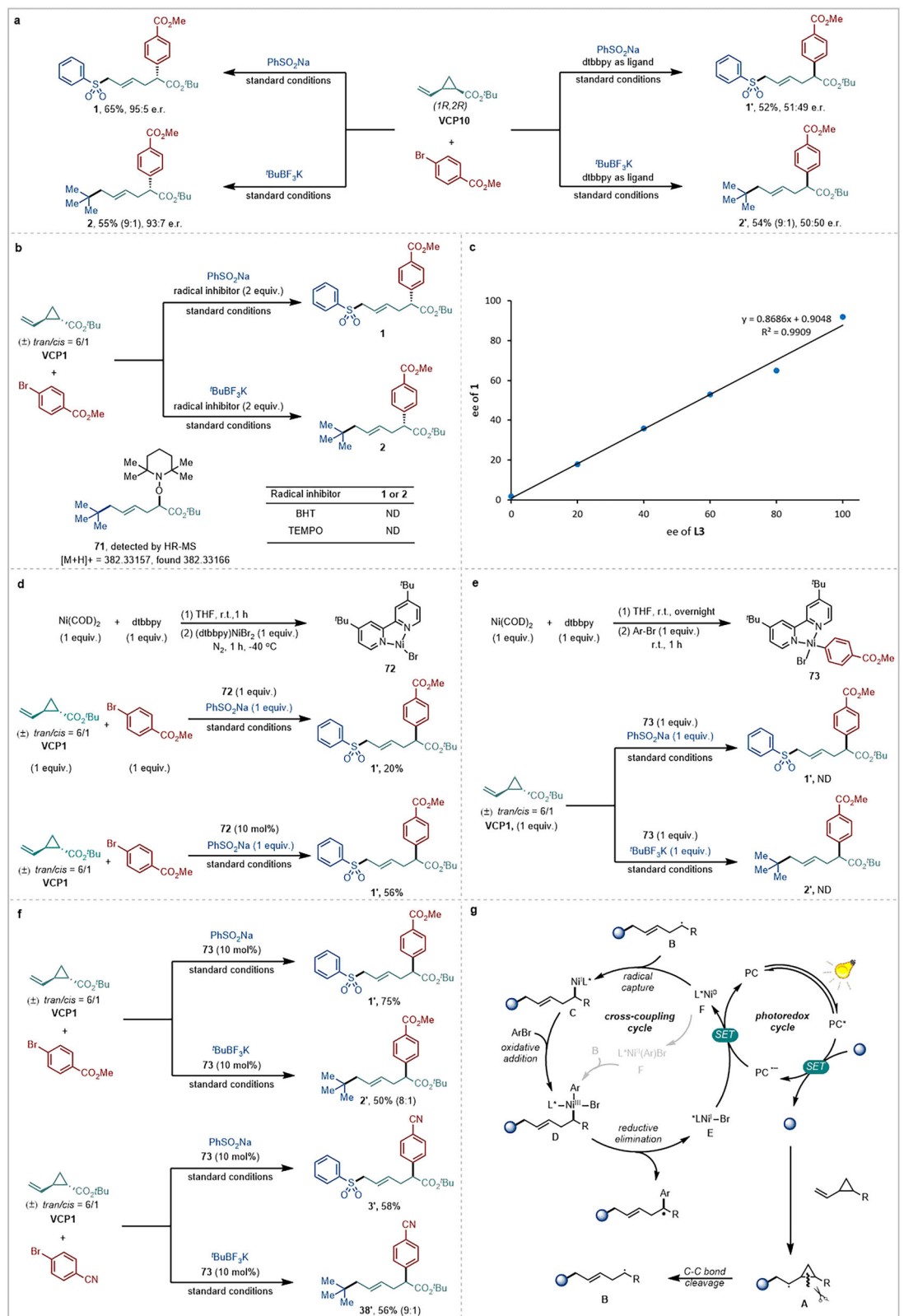

**Fig. 5 | Mechanistic investigations. a** Substrate control experiments. **b** Radical inhibition experiments. **c** Non-linear effect experiment. **d** Stoichiometric and catalytic experiments with dtbbpyNi(I)Br complex. **e** Stoichiometric experiments with ArNi(II)Br complex. **f** Catalytic and crossover experiments with ArNi(II)Br complex. **g** Proposed catalytic cycle.

bromide (If solid, 0.1 mmol, 1 equiv.), Ir[dF(CF$_3$)ppy)]$_2$(bpy)PF$_6$ (3 mol%), NiCl$_2$·Py$_4$ (10 mol %) and **L2** (15 mol%) and then introduced into a nitrogen-filled glovebox. Dry THF (2 mL) was added, then the reaction vessel was then capped and removed from the glovebox. Vinyl

cyclopropane (0.2 mmol, 2.0 equiv.) and aryl or alkenyl bromide (if liquid, 0.1 mmol, 1 equiv.) were subsequently added. The reaction was stirred (800 rpm) under irradiation with a 427 nm 45 W Kessil LED at room temperature for 24 h. The reaction was quenched with

saturated aq. NaCl (1 mL) and the resulting mixture were extracted with EtOAc (3 × 2 mL). The organic phase was concentrated under reduced pressure, and the residue purified by column chromatography on silica gel. The enantiomeric ratio (e.r.) and the *E/Z* ratio of the product were determined by HPLC with a chiral stationary phase.

## Data availability

The authors declare that the data supporting the findings of this study are available within the paper and its supplementary information files. Crystallographic data for the structures reported in this Article have been deposited at the Cambridge Crystallographic Data Centre, under deposition numbers CCDC 2421141 (**60**). Copies of the data can be obtained free of charge via https://www.ccdc.cam.ac.uk/structures/. All data are available from the corresponding author upon request.

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

## Acknowledgements

We thank Dr. Olivier Blacque for the X-ray diffraction analysis of **60**. This publication was created as part of NCCR Catalysis, a National Centre of Competence in Research funded by the Swiss National Science Foundation. The Swiss National Science Foundation (SNF es 200021_184986/1) is also acknowledged for financial support. The Forschungskredit of the University of Zurich is also acknowledged for support to M.E.L.

## Author contributions

C.N. and X.D. conceived the project. X.D., M.E.L., and G.K. performed the experiments and analyzed the data. All the authors participated in the preparation of the manuscript.

## Competing interests

The authors declare no competing interests.
