## [Transparent Peer Review file · Nature Communications]

Vinyl Cyclopropanes: a Unifying Platform for Enantioselective Remote Difunctionalization of Alkenes

Corresponding Author: Professor Cristina Nevado

Version 0:

Reviewer comments:

Reviewer #1

(Remarks to the Author)

This paper reports the asymmetric 1,5-carbosulfonylation and 1,5-dicarbofunctionalization of vinyl cyclopropanes via a radical addition–C–C bond cleavage/cross-coupling cascade by nickel/photoredox catalysis. Specifically, the authors performed two reactions including 1,5-carbosulfonylation and 1,5-dicarbofunctionalization. The mechanism proceeds through radical addition followed by ring opening of the cyclopropyl ring to generate a radical that is captured by the nickel catalyst. This Nickel catalysis is well-established and well-reported mechanism. Therefore, the current work should be expanded to the more diverse scope. Also, the Ni-alkyl intermediate in the proposed mechanism should be discussed better to explain the role of carbonyl groups because all cyclopropanes contain a carbonyl group. This work can be reconsidered for Nature Commun after revisions.

Comments

-The ester group of vinyl cyclopropane seems to be a prerequisite for this reaction, which limits the substrate scope. Can the authors expand the scope of VCP

-One of the issues is the ee for most of the substrates which are around 80% ee. The reviewer thinks that this is not sufficient for a highly efficient asymmetric reaction.

-Alkyl halides (bromide or iodide) can be utilized instead of aryl bromide? (Csp³-Csp³ coupling)

-The authors have prepared the Ni(II) complex (67) and found that it is not the active species. Include control experiments with the Ni(I) complex. If it cannot be isolated, in-situ preparation may be acceptable.

-Discuss whether 1,5-HAT is observed when the radical was generated after C–C bond cleavage.

-15-Crown-5 and B₂Pin₂ in the 1,5-carbosulfonylation are essential?

Reviewer #2

(Remarks to the Author)

In this manuscript, Nevado and colleagues report the first nickel- and photoredox dual-catalyzed asymmetric remote 1,5-carbosulfonylation and 1,5-dicarbofunctionalization of vinyl cyclopropanes, achieving high yields and excellent enantioselectivities. A broad substrate scope was explored, and notably, the synthetic utility was demonstrated through late-stage functionalization of complex molecules and further derivatization of the products, yielding highly promising results. The origin of enantioselectivity was investigated through control experiments.

The manuscript is well-written and scientifically rigorous. Overall, this work represents a significant advancement and would be of great interest to the broad readership of Nature Communications. Therefore, I strongly recommend its publication in Nature Communications following minor revisions. Some comments are listed as following:

Main concerns:

Main Concerns:

1. The authors should include a photographic representation of the experimental setup in the Supporting Information.
2. What are the outcomes when using other aryl halide coupling partners, such as methyl 4-iodobenzoate?

3. Recent reports have observed a non-linear effect in similar reactions. Could the authors discuss the potential for a non-linear effect in this asymmetric reaction? Addressing this could provide valuable insights into the reaction mechanism and stereoselectivity.
 4. What role does B_2Pin_2 play in the reaction? This additive is not commonly used—could the authors provide an explanation? Clarifying its function would be highly beneficial for understanding the mechanism.
 5. The authors should specify how the E/Z ratio is determined.
 6. For late-stage functionalization of complex molecules, how is the dr value determined in the absence of an HPLC trace?
- Minor Points:
7. Some references are missing page and volume numbers, such as Ref. 19.
 8. In Table S1 of the Supporting Information, the structures of L2 and L3 appear identical. The authors should revise L3 by changing sBu to iPr.
 9. In Table S7, it should be “instead” and “solvent”

Reviewer #3

(Remarks to the Author)

The manuscript describes a nickel/photoredox dual-catalyzed asymmetric remote 1,5-carbosulfonylation and 1,5-dicarbofunctionalization of vinyl cyclopropanes. However, a similar study by Wang et al. (Chin. J. Chem. 2025, 43, 1271–1278; DOI: 10.1002/cjoc.202500074) reports a nickel-catalyzed asymmetric reductive 1,4- and 1,5-dicarbofunctionalization, which raises concerns regarding the novelty of this work. Therefore, I recommend rejecting the manuscript for publication in Nature Communications.

Version 1:

Reviewer comments:

Reviewer #1

(Remarks to the Author)

To address the issues and questions raised previously, the authors have conducted additional experimental work to expand the scope and provide clarification in the revision. Although a recently published paper (Chin. J. Chem. 2025, 43, 1271) shares conceptual similarities with the current work, which somewhat diminishes the novelty of this study, this paper was published during the revision process and therefore does not appear to pose a significant concern.

Reviewer #2

(Remarks to the Author)

The authors have nicely addressed all the concerns of this reviewer. I recommend publishing it as it is.

COMMENTS TO AUTHOR:

Reviewer 1:

1. The ester group of vinyl cyclopropane seems to be a prerequisite for this reaction, which limits the substrate scope. Can the authors expand the scope of VCP.

As shown below, additional examples of VCPs without ester groups have been incorporated into both the main text and the SI of our manuscript. In particular, 1-phenyl-2-vinylcyclopropane could be successfully engaged under the reaction conditions to give the corresponding products with moderate yields and excellent levels of absolute stereocontrol. Moreover, methyl-substituted substrates furnished the desired remote difunctionalization products as single geometric isomers in excellent yields and enantioselectivities.

2. One of the issues is the ee for most of the substrates which are around 80% ee. The reviewer thinks that this is not sufficient for a highly efficient asymmetric reaction.

As shown above, 1-phenyl-2-vinylcyclopropane and its derivatives gave the remote carbosulfonylation products in excellent enantioselectivities (up to 99:1 er). In addition, in the case of product **2**, the e.r. could be increased from 93:7 to 96:4 by employing **L3** as ligand, albeit with a slight decrease in yield. This modification also works very well with other substrates: remote dicarbofunctionalization products **42**, **44**, **48**, **52** and **54** were obtained with increased er's with this modified protocol. These additional examples have been summarized below and can be now found both in the main text and SI of our manuscript.

2, 62% (9:1), 93:7 e.r.
 46% (9:1), 96:4 e.r.^a

42, 41% (8:1), 90:10 e.r.
 38% (8:1), 94:6 e.r.^{a,b}

44, 67% (7:1), 91:9 e.r.
 55% (6:1), 95:5 e.r.^{a,b}

48, 57% (8:1), 90:10 e.r.
 52% (9:1), 94:6 e.r.^{a,b}

52, 42% (9:1), 91:9 e.r.
 35% (9:1), 94:6 e.r.^{a,b}

54, 57% (7:1), 91:9 e.r.
 50% (9:1), 94:6 e.r.^{a,b}

^a L3 as ligand. ^b [0.025 M]

3. Alkyl halides (bromide or iodide) can be utilized instead of aryl bromide? (Csp³-Csp³ coupling).

Several alkyl halides were submitted to the standard reaction conditions with each radical precursor type (arenesulfinate or alkyl trifluoroborate). In each case, the Csp³-Csp³ coupling product was not observed by NMR or HRMS. Study of the reaction outcomes with arenesulfinate radical precursors gave us some insight into why the reaction failed. Benzylic alkyl bromides gave as the major product the sulfone resulting from direct electrophilic alkylation of the sulfinate. Conversely, secondary alkyl bromides gave clean conversion to the hydrosulfonylation product. The fact that Csp³-Csp³ coupling was not competitive in either case indicates that at least one of the elementary steps to incorporate the alkyl halide (oxidative addition or reductive elimination) is prohibitively slow in our system.

nd

nd

nd

nd

4. The authors have prepared the Ni(II) complex (67) and found that it is not the active species. Include control experiments with the Ni(I) complex. If it cannot be isolated, in-situ preparation may be acceptable.

The suggested Ni(I) complex has been synthesized using the procedure shown below. Additional control experiments have been carried out to complement the existing ones supporting the proposed mechanism for this transformation. As shown below, when Ni(I)-Br complex **72** was used in a stoichiometric fashion in the reaction of **VCP1** and sodium benzenesulfinate, the corresponding remote arylsulfonated product **1'** could be obtained in 20% yield. Further, using catalytic amounts of complex **72** generated the desired product **1'** in 56% yield. This result implicates Ni(I) species as possible active catalysts, and supports a Ni(0)-Ni(I)-Ni(III) catalytic cycle. These results have now been incorporated in the main text and the SI of our manuscript.

5. Discuss whether 1,5-HAT is observed when the radical was generated after C-C bond cleavage.

We have never observed products resulting from 1,5-HAT after the C-C bond cleavage. We believe that this is in part because the cyclopropane ring-opening step generates predominantly *E*-configured olefins, which are geometrically constrained such that 1,5-HAT is not possible. In the case of alkyl trifluoroborate radical precursors, which produce more appreciable amounts of *Z*-configured olefins, it is likely that conformations permitting 1,5-HAT are prohibited by steric clash between the bulky tertiary alkylic groups at each end of the molecule.

6. 15-Crown-5 and B₂Pin₂ in the 1,5-carbosulfonylation are essential?

15-Crown-5 is essential to solubilize the sodium arenesulfinate radical precursor in our optimal solvent system. When polar aprotic solvents (e.g., DMSO) are employed, high reactivity may be observed even in the absence of 15-crown-5, but such regimes increase the rate of racemic background reactions (presumably by ligating Ni(II) to form an achiral precatalyst) and therefore harm the enantioselectivity of the reaction.

B₂pin₂ may be omitted at the cost of diminished yield (see optimization section of supporting information). Its role is in generating the low-valent nickel species which are the active catalyst in the reaction. This role has strong precedence in the field of nickel catalysis (see, for example: *Adv. Sci.* **2024**, *11*, 2404301). In the absence of B₂pin₂, other species (e.g., sulfinate salts) must fill the role of terminal reductant in the induction period. We surmise that these alternative processes are less efficient than the well-known reduction with B₂pin₂, and so the overall efficiency of the reaction suffers as a result.

Reviewer 2:

1. The authors should include a photographic representation of the experimental setup in the Supporting Information.

Thank you for the suggestion. The photographs of the reaction setup have been added in SI as Figure S1.

2. What are the outcomes when using other aryl halide coupling partners, such as methyl 4-iodobenzoate?

Methyl 4-iodobenzoate and 4-iodobenzonitrile were successfully incorporated as reaction partners in the remote carbosulfonylation reaction, albeit with a slight decrease in both yields and enantioselectivities compared to the corresponding aryl bromides. Furthermore, methyl 4-iodobenzoate was also subjected to the standard reaction conditions for remote dicarbofunctionalization and the corresponding product could be obtained with comparable enantioselectivity to that observed with the aryl bromide, albeit in only moderate yield.

3. Recent reports have observed a non-linear effect in similar reactions. Could the authors discuss the potential for a non-linear effect in this asymmetric reaction? Addressing this could provide valuable insights into the reaction mechanism and stereoselectivity.

As suggested, a series of Ni and photoredox dual catalyzed asymmetric remote dicarbofunctionalization were conducted, varying the enantiopurity of the ligand (L3) under the standard reaction conditions. A linear effect was observed on the asymmetric induction, indicating a 1:1 binding pattern between L3 and Ni in the active catalytic species.

4. What role does B_2Pin_2 play in the reaction? This additive is not commonly used—could the authors provide an explanation? Clarifying its function would be highly beneficial for understanding the mechanism.

See response to reviewer 1

5. The authors should specify how the E/Z ratio is determined.

The E/Z ratio is determined by HPLC, as noted in the relevant figure captions of the manuscript.

6. For late-stage functionalization of complex molecules, how is the dr value determined in the absence of an HPLC trace?

The d.r. is determined in these cases by 1H NMR spectroscopy. This will be noted in the relevant figure caption of the revised manuscript.

7. Some references are missing page and volume numbers, such as Ref. 19.

Thanks for the reviewer's suggestion. We have revised the references accordingly.

8. In Table S1 of the Supporting Information, the structures of L2 and L3 appear identical. The authors should revise L3 by changing sBu to iPr.

Thanks for the reviewer's suggestion. We have revised this part according to the reviewer's suggestion.

9. In Table S7, it should be “instead” and “solvent”

Thanks for the reviewer's suggestion. We have revised this part according to the reviewer's suggestion.

Reviewer 3:

The manuscript describes a nickel/photoredox dual-catalyzed asymmetric remote 1,5-carbosulfonylation and 1,5-dicarbonylation of vinyl cyclopropanes. However, a similar study by Wang et al. (Chin. J. Chem. 2025, 43, 1271–1278; DOI: 10.1002/cjoc.202500074) reports a nickel-

catalyzed asymmetric reductive 1,4- and 1,5-dicarbofunctionalization, which raises concerns regarding the novelty of this work. Therefore, I recommend rejecting the manuscript for publication in Nature Communications.

As discussed in the manuscript, vinylcyclopropanes have been employed in the past for remote asymmetric difunctionalization in a more limited context, as a part of a broader work (typically with few examples, low yields and/or low enantioselectivity). However, our work is the first to describe a *general* platform to achieve both remote dicarbofunctionalization and remote hetero-carbofunctionalization in asymmetric fashion. This generality with respect to radical precursor is made possible in part by the use of a redox-neutral regime with no (super)stoichiometric oxidants or reductants. Moreover, the present work is general with respect to vinylcyclopropane scaffolds: we have shown that our system is competent in the presence of various attached functional groups (ester, arene, amide, sulfone, ketone) and substitution patterns at both the vinyl and cyclopropane moieties. This unprecedented generality, along with its high efficiency, enantioselectivity and functional group tolerance, sets the present work apart from the piecemeal contributions found in previous reports. We have added this contribution, which was published during the revision process, as ref. 48 in our manuscript.